

# Gauged sigma models and magnetic Skyrmions

**Bernd J. Schroers**

Maxwell Institute for Mathematical Sciences and Department of Mathematics,
Heriot-Watt University, Edinburgh EH14 4AS, UK

b.j.schroers@hw.ac.uk

## Abstract

We define a gauged non-linear sigma model for a 2-sphere valued field and a $SU(2)$ connection on an arbitrary Riemann surface whose energy functional reduces to that for critically coupled magnetic skyrmions in the plane, with arbitrary Dzyaloshinskii-Moriya interaction, for a suitably chosen gauge field. We use the interplay of unitary and holomorphic structures to derive a general solution of the first order Bogomol'nyi equation of the model for any given connection. We illustrate this formula with examples, and also point out applications to the study of impurities.



# 1   Introduction

This paper has two goals. The first is to show that a large class of models for static magnetic skyrmions, which includes the most general form of the Dzyaloshinskii-Moriya (DM) interaction [1,2], can be formulated as a (non-abelian) gauged non-linear sigma model and solved explicitly. The second is to study the relevant gauged sigma models more generally and to explain the interesting differential geometry which lies behind their solvability.

Magnetic skyrmions [3] represent the most recent and perhaps the richest application of the concept of topological solitons to condensed matter physics and, potentially, the technology of future magnetic information storage [4]. There is a vast and rapidly growing literature on the subject, partly reviewed in [5] and, with a particular emphasis on DM terms in the notation which we will adopt, in [6]. However, it was only noticed recently in [7] that, among the large class of mathematical models for magnetic skyrmions, there are some, called critically coupled in [7], where an infinite family of skyrmion configurations can be found exactly. These configurations satisfy so-called Bogomol'nyi equations and may be viewed as generalisations of the Belavin-Polyakov self-dual solitons [8].

Our discussion here is an extension and generalisation of the paper [7], both in terms of the range of applications and in terms of the underlying mathematics. Instead of the two-parameter family of DM terms considered in [7] we treat the most general DM term and point out an application to the study of impurities. At the same time, we clarify the underlying mathematics by defining the relevant gauged non-linear sigma model in its most general, geometrically natural form. This leads to a model which we believe to be of independent interest. It involves a $S^2$-valued field defined on an arbitrary Riemann surface and coupled to a fixed, non-abelian connection

There is a large literature on gauged non-linear sigma models, but as far as we know the model and energy functonal we study here has not been considered before. The solutions of our model obey Bogomol'nyi equations which are similar to those occurring in the abelian non-linear sigma model of [9] and its generalisations (see e.g. the book [10]) and also in the non-abelian model considered by Nardelli in [11]. However, our energy functional is different, and our non-abelian gauge field plays a different role, namely that of an arbitrary but fixed background.

Magnetic skyrmions are mathematically described as topological solitons in the magnetisation field $\boldsymbol{n}$ of magnetic materials. The latter is a map from a surface $\Sigma$ to the 2-sphere $S^2$. The energy expression for the magnetisation field $\boldsymbol{n}$ typically involves a Heisenberg or Dirichlet term (quadratic in derivatives) the crucial DM term (linear in derivatives) and various potential terms (no derivatives). To be specific, consider the most general form of such a model in the plane $\Sigma = \mathbb{R}^2$. The energy functional is

$$E[\boldsymbol{n}] = \int_{\mathbb{R}^2} \frac{1}{2}(\nabla \boldsymbol{n})^2 + \sum_{a=1}^{3}\sum_{i=1}^{2} \mathcal{D}_{ai}[\partial_i \boldsymbol{n} \times \boldsymbol{n}]_a + V(\boldsymbol{n})\, dx_1 dx_2, \tag{1}$$

where $V$ is a potential which includes a Zeeman term and anisotropy terms, and $\mathcal{D}$ is the tensor parametrising the DM interaction. It is sometimes called spiralization tensor, see for example the paper [6], to which we also refer the reader for a discussion of the associated physics and examples. In order to analyse such a model, it is useful to clarify the mathematical structures which enter its definition.

In two dimensions, the Dirichlet energy expression only depends on a conformal structure of the domain $\Sigma$. However, the potential terms require an integration measure, so that one would expect the full model to depend on a metric structure on $\Sigma$. In any case, the energy expression makes use of the (standard) metric on the target 2-sphere.

The reformulation of the model (1) in [7] for a particular potential and DM term as a gauged non-linear sigma model only requires a conformal structure on $\Sigma$ and a choice of an $SU(2)$ connection. However, its solution in terms of holomorphic data makes essential use of the complex structure on the target, i.e. of the identification of the 2-sphere with the complex projective line $\mathbb{C}P^1$. This suggests a more general formulation of the gauged non-linear sigma model on a Riemann surface $\Sigma$ and also a more general understanding of its solution in terms of the interplay between the metric and complex structure on the target 2-sphere.

In this paper, we give such a formulation and show that the resulting model can always, at least locally, be solved in terms of an $SL(2,\mathbb{C})$-valued map which relates holomorphic and unitary gauges for a $\mathbb{C}P^1$-bundle over $\Sigma$. A similar interplay between unitary and holomorphic structures is much studied in the context of self-dual connections on 4-manifolds [12,13], but the 2-dimensional version which we need here is the simplest example of a general theory of connections, often called Chern connections, which are compatible with unitary and holomorphic structures on a complex vector bundle, see [14].

Our presentation is organised as follows. We begin, in Sect. 2, with a definition and discussion of the gauged non-linear sigma model in the most general setting of an arbitrary Riemann surface $\Sigma$. We define the energy functional and derive a lower bound for it in terms of a topological invariant and the boundary behaviour of the magnetisation field (provided $\Sigma$ has a boundary). We then derive a first order Bogomol'nyi equation for configurations which saturate this bound. In Sect. 3 we begin with a brief review of relevant complex geometry, in particular the concept of a Chern connection, and then derive a general formula for solutions of the Bogomol'nyi equation for any given $SU(2)$ connection. In Sect. 4 we apply the results of Sects. 2 and 3 to magnetic skyrmions by writing the energy functional (1) for a particular choice of potential (associated to the spiralization tensor $\mathcal{D}$) as a gauged non-linear sigma model. This requires a translation of the spiralization tensor $\mathcal{D}$ into a $SU(2)$ connection, which we shall explain. We use the general formula of Sect. 3 to obtain infinite families of exact magnetic skyrmions for any given spiralization tensor and an associated potential, and discuss some examples. We also explain how our formalism can be used to define and solve models of topological solitons interacting with impurities. Our final Sect. 5 contains a conclusion and an outlook.

The logic of our mathematical derivation requires that we begin our discussion with gauged non-linear sigma models, but readers who are predominantly interested in magnetic skyrmions and willing to take the solution formula on trust are invited to skip to Sect. 4.

## 2  Gauged sigma models on a Riemann surface

### 2.1  Conventions

The gauged sigma model we want to consider can be defined on any Riemann surface $\Sigma$, i.e. on any one-dimensional complex manifold, with our without boundary. We will define the model using invariant notation, but for concrete and explicit expressions we use a local complex coordinate $z = x_1 + ix_2$. We also use the standard notation

$$dz = dx_1 + ix_2, \qquad \partial_z = \frac{1}{2}(\partial_1 - i\partial_2), \quad \partial_{\bar{z}} = \frac{1}{2}(\partial_1 + i\partial_2). \tag{2}$$

The Hodge $\star$ operator on 1-forms is determined by the complex structure; in local coordinates it is

$$\star dz = dx_2 - idx_1 = -idz, \qquad \star d\bar{z} = dx_2 + idx_1 = id\bar{z}. \tag{3}$$

Note also that, for any two 1-forms $\alpha, \beta$ on $\Sigma$,

$$\star \star \alpha = -\alpha \quad \text{and} \quad \alpha \wedge \star\beta = \beta \wedge \star\alpha. \tag{4}$$

Consider now a principal $SU(2)$-bundle $P$ over $\Sigma$ with a connection, and the associated adjoint bundle as well as the unit 2-sphere bundle $P \times^{\mathrm{Ad}} S^2$ in the adjoint bundle. We think of the fibre $S^2$ as the round 2-sphere of radius 1 inside the Lie algebra $su(2)$, with $SU(2)$ acting in the adjoint representation. Locally, in an open set $U \subset \Sigma$, a section of $P \times^{\mathrm{Ad}} S^2$ is a map

$$n : U \to S^2 \subset su(2), \tag{5}$$

and the connection is given by a $su(2)$-valued 1-form $A$ on $U$. The exterior covariant derivative of $n$ and the curvature is given by the usual expressions

$$Dn = dn + [A, n], \quad F = dA + \frac{1}{2}[A, A]. \tag{6}$$

Here and in the rest of the paper we suppress the wedge product in the commutator of Lie algebra-valued 1-forms. Gauge transformations are determined locally by functions $u : U \to SU(2)$ and take the form

$$n \to unu^{-1}, \qquad A \to uAu^{-1} + udu^{-1}. \tag{7}$$

In local coordinates on $\Sigma$, we expand

$$A = A_1 dx_1 + A_2 dx_2 = A_z dz + A_{\bar{z}} d\bar{z}, \tag{8}$$

where we defined $A_z = \frac{1}{2}(A_1 - iA_2)$, $A_{\bar{z}} = \frac{1}{2}(A_1 + iA_2)$, and similarly for the curvature

$$F = F_{12} dx_1 \wedge dx_2, \qquad F_{12} = \partial_1 A_2 - \partial_2 A_1 + [A_1, A_2]. \tag{9}$$

We use $su(2)$ generators $t_a = -\frac{i}{2}\tau_a$ , $a = 1, 2, 3$, where $\tau_a$ are the Pauli matrices, which are normalised so that $[t_1, t_2] = t_3 +$ cycl., and also define raising and lowering operators

$$t_+ = t_1 + it_2, \quad t_- = t_1 - it_2. \tag{10}$$

These naturally lie in the complexified $su(2)$ Lie algebra, so in $sl(2, \mathbb{C})$. For the inner product of $m, n \in su(2)$ we use a re-scaled trace and write

$$(m, n) = -2\mathrm{tr}(mn), \tag{11}$$

as well as $|n|^2 = (n, n)$. Our normalisation is such that $(t_a, t_b) = \delta_{ab}$.

## 2.2 Energy and variational equations

The energy functional defining the gauged sigma model we study in this paper is

$$E[A, n] = \frac{1}{2}\int_{\Sigma}(Dn, \wedge \star Dn) - \int_{\Sigma}(F, n), \tag{12}$$

or, in local coordinates and in terms of $D_i = \partial_i + [A_i, \cdot]$, $i = 1, 2$,

$$E[A, n] = \int_{\Sigma}\left(\frac{1}{2}|D_1 n|^2 + \frac{1}{2}|D_2 n|^2 - (F_{12}, n)\right) dx_1 \wedge dx_2. \tag{13}$$

Clearly, the covariant Dirichlet term depends on the complex structure (through $\star$), but the curvature term does not and is topological in that sense. The energy is manifestly invariant under the gauge transformations (7).

As our notation indicates, we think of the energy as a functional of the connection $A$ and the section $n$. Postponing a discussion of boundary terms we initially assume that the Riemann surface $\Sigma$ has no boundary. Then variation with respect to the connection gives

$$\delta E = \int_\Sigma (Dn, \wedge \star [\delta A, n]) - (n, (d\delta A + [\delta A, A])) = \int_\Sigma (\delta A, \wedge \star [n, Dn]) - (\delta A, \wedge Dn). \quad (14)$$

Setting this to zero for all $\delta A$ gives the Euler-Lagrange equation

$$\star [n, Dn] = Dn. \quad (15)$$

Using (3), this can also be written as

$$D_{\bar z} n = i[n, D_{\bar z} n] \Longleftrightarrow D_z n = -i[n, D_z n]. \quad (16)$$

Since $[n, \cdot]$ is the complex structure in the cotangent space to $S^2$ at $n$, the equation (15) is a holomorphicity condition. This will become more obvious and useful when we study this equation in complex coordinates for $S^2$ in Sect. 3.2.

The variation with respect to $n$, using $\delta n = [\epsilon, n]$ to preserve $|n| = 1$ and neglecting boundary terms gives

$$\delta E = \int_\Sigma (D\delta n, \wedge \star Dn) - (\delta n, F) = -\int_\Sigma (\delta n, (D \star Dn + F)) = -\int_\Sigma (\epsilon, [n, D \star Dn + F]). \quad (17)$$

Setting this to zero for all $\epsilon$ leads to the variational equation

$$[n, D \star Dn + F] = 0. \quad (18)$$

It is not difficult to check that the first order equation (15) actually implies the second order equation (18). Applying $\star$ to (15) and differentiating, we obtain

$$D \star Dn = -D[n, Dn] = -[Dn, Dn] - [n, D^2 n] = -[Dn, Dn] - [n, [F, n]]. \quad (19)$$

Now we use that $[Dn, Dn]$ is in the direction of $n$ to deduce

$$[n, D \star Dn] = -[n, [n[F, n]]] = -[n, F], \quad (20)$$

as claimed. The equation (15) is therefore the only equation we need to consider. We now show that it can also be interpreted as a Bogomol'nyi equation in this model.

## 2.3 The Bogomol'nyi equation

The logical dependence of the two variational equations can be understood better by noting that (15) can also be obtained via a Bogomol'nyi argument. To show this, we need 't Hooft's identity [15] relating the integrand for the degree,

$$4\pi \deg[n] = \int_\Sigma \frac{1}{2}(n, [dn, dn]), \quad (21)$$

to its covariant version:

$$\frac{1}{2}(n, [Dn, Dn]) = \frac{1}{2}(n, [dn, dn]) + (F, n) - d(A, n). \quad (22)$$

Now use (4) and the cyclical property of the triple product to note

$$((Dn - \star[n, Dn]), \wedge \star (Dn - \star[n, Dn])) = 2(Dn, \wedge \star Dn) - 2(n, [Dn, Dn]). \quad (23)$$

This allows us to write the energy as

$$E[A, n] = \frac{1}{4} \int_\Sigma ((Dn - \star[n, Dn]), \wedge \star (Dn - \star[n, Dn])) + \frac{1}{2} \int_\Sigma (n, [Dn, Dn]) - \int_\Sigma (F, n). \quad (24)$$

Combining this with the identity (22), we deduce

$$E[A, n] = \frac{1}{4} \int_\Sigma ((Dn - \star[n, Dn]), \wedge \star (Dn - \star[n, Dn])) + \frac{1}{2} \int_\Sigma (n, [dn, dn]) - \int_{\partial\Sigma} (A, n), \quad (25)$$

with the last term of course vanishing when $\Sigma$ has no boundary. We conclude that the energy is bounded below by terms which only depend on topology (the degree of $n$) or on boundary behaviour (if there is a boundary). If both are kept fixed, the energy is minimised iff the first order equation (15) holds. The energy of such Bogomol'nyi configurations is

$$E_B[A, n] = \frac{1}{2} \int_\Sigma (n, [dn, dn]) - \int_{\partial\Sigma} (A, n). \quad (26)$$

The equation (15) is thus seen to be a Bogomol'nyi equation in the general sense of characterising minima of energy functionals subject to topological or boundary conditions [16]. Such equations usually imply the variational equations. This is the case here, too, but in a somewhat unusual way. The Bogomol'nyi equation of the model *is* the variational equation (15) with respect to the connection, and implies the second order variational equation with respect to the field $n$, as we already showed.

The Bogomol'nyi equation (15) clearly does not uniquely determine both the connection and the section $n$. It is easy to write down infinitely many solutions for the connection $A$ for *any* given smooth section $n$ in the form

$$A = an - (p[n, dn] + q \star dn) + r(\star[n, dn] + dn), \quad (27)$$

where $a$ is a 1-form and $p, q, r$ are real functions, with $r$ arbitrary but $p, q$ satisfying $p + q = 1$. The choice $A = 0$ is possible when $\star[n, dn] = dn$, and is then realised with $p = q = \frac{1}{2}$, and $a = 0, r = 0$. This is expected because in that case $n$ satisfies the ungauged Bogomol'nyi equation.

Note that the 1-form $[n, dn]$ is naturally associated to $n$ as the Levi-Civita connection on the plane bundle orthogonal to $n$ inside the trivial bundle $\Sigma \times su(2)$, and that the remaining terms other than $an$ are obtained from the Levi-Civita 1-form by applying the complex structures $\star$ on the domain or $[n, \cdot]$ on the target. Also note that gauge transformations $g = \exp(\alpha n)$ which fix $n$ (so $\alpha$ is a function) determine gauge equivalent solutions for fixed $n$.

Alternatively, one can use transformations (7) to rotate $n$ in a fixed direction (say $t_3$) in the Lie algebra on some open set $U \in \Sigma$. In this gauge, $A$ is determined by the algebraic condition

$$\star[t_3, [A, t_3]] = [A, t_3]. \quad (28)$$

Solving for $A$ when $n$ is given may be of interest in some applications, e.g. when one would like a particular configuration to be a solution in the presence of an impurity, see our discussion in Sect. 4.4. However, we will now focus on the opposite situation where $A$ is a given background gauge field, and we solve (15) for $n$ in this background.

## 2.4 Boundary terms

So far we have ignored boundary terms which arise in the derivation of the variational equations for the functional (12). When $\Sigma$ does have a boundary $\partial\Sigma$, boundary terms will generally

only vanish if we impose a suitable boundary condition. When $\Sigma$ is an open set - such as $\mathbb{C}$, or the upper half-plane - we need to impose suitable fall-off conditions as we approach 'infinity'. In order to discuss these matters in any detail we would need to fix the surface $\Sigma$ and the background gauge field $A$ we want to consider. We will not do this here, but make some general observations.

The boundary term which arises in the variation (17) of (12) with respect to $n$ is

$$\int_{\partial\Sigma} (\delta n, \star Dn) = \int_{\partial\Sigma} (\epsilon, \star[n, Dn]). \tag{29}$$

If $\partial\Sigma$ is an actual boundary and we impose a Dirichlet boundary condition $n_{|\partial\Sigma} = n_\infty$ this term vanishes because we must require $\delta n = [\epsilon, n] = 0$ on the boundary. However, when $\Sigma = \mathbb{C}$, the requirements $\lim_{|z|\to\infty} n(z) = n_\infty$ and $\lim_{|z|\to\infty} \epsilon(z) = 0$ are not sufficient to ensure the vanishing or even well-definedness of the integral (29): when the gauge potential $A$ does not vanish in the limit $|z| \to \infty$ the term in the integrand of (29) containing $A$ may have a non-zero integral even when $\epsilon$ vanishes in this limit. The situation can be improved by considering the modified energy functional

$$\tilde{E}[A, n] = \int_\Sigma \frac{1}{2}(Dn, \wedge \star Dn) - (F, n) + \int_{\partial\Sigma} (A, n). \tag{30}$$

The inclusion of the boundary term means that the modified energy is bounded below by the integral defining the degree, see (25). Now variation with respect to $n$ gives

$$\delta\tilde{E} = -\int_\Sigma (\delta n, (D \star Dn + F)) + \int_{\partial\Sigma} (\delta n, (\star Dn + A)). \tag{31}$$

Using again $\delta n = [\epsilon, n]$ we only need to assume the Bogomol'nyi equation (15) on the boundary to obtain

$$\delta\tilde{E} = -\int_\Sigma (\delta n, (D \star Dn + F)) + \int_{\partial\Sigma} (\epsilon, dn). \tag{32}$$

Comparing the new boundary term with (29) we observe that the modification of the energy expression has removed the troublesome term in (29) involving the gauge potential.

In summary, the modified energy functional (30) is a better starting point for a well-defined variational problem in the presence of a boundary, though the details will depend on the surface and boundary under consideration. Requiring the Bogomol'nyi equation in the boundary region and mild fall-off conditions for $\epsilon$ and and $dn$ are sufficient to remove boundary terms which arise in the variation. The bulk equations for $n$ are, of course, the equations (18) already derived in the absence of a boundary.

## 3 Solving the Bogomol'nyi equation

### 3.1 Holomorphic versus unitary structures

We will now show how to solve the Bogomol'nyi equation (15) for a given connection. The idea is to exploit the interplay between a holomorphic and a unitary structure on a complex vector bundle over $\Sigma$, and the special properties of the unique connection, often called the Chern connection, which is compatible with both. The underlying theory is covered, for example in [13, 14] and also more informally in [12].

We consider only the setting which is relevant for our discussion, so look at holomorphic $\mathbb{C}^2$-bundles over $\Sigma$ with a unitary structure (a Hermitian inner product on the fibres). Any

such vector bundle, denoted $E$ in the following, has an associated projective bundle; this is a holomorphic $\mathbb{C}P^1$-bundle and, with the unitary structure, will be identified with the $S^2$-bundle $P \times^{\mathrm{Ad}} S^2$ of Sect. 2.1. We use the standard notation of $\partial$ and $\bar{\partial}$ for the exterior derivative followed by projection onto differential forms of type $(1,0)$ and $(0,1)$ on $\Sigma$, so in our local coordinates and applied to functions $f$,

$$\partial f = \partial_z f \, dz, \qquad \bar{\partial} f = \partial_{\bar{z}} f \, d\bar{z}. \tag{33}$$

Now consider a connection on the vector bundle $E$. The associated covariant derivative $D = d + A$ can be split into

$$\partial_A = \partial + A_z dz, \qquad \bar{\partial}_A = \bar{\partial} + A_{\bar{z}} d\bar{z}, \tag{34}$$

where $A = A_z dz + A_{\bar{z}} d\bar{z}$ is simply the split (8) of the matrix-valued 1-form $A$ into forms of type $(1,0)$ and $(0,1)$. Such a connection is called unitary if it preserves the Hermitian inner product on the fibres, and it is called compatible with the holomorphic structure of $E$ if $\bar{\partial}_A \vec{w} = 0$ for every holomorphic section[1] $\vec{w}$ of $E$. If a connection is compatible with both structures then, in a unitary gauge (a local choice of an orthonormal basis of the fibre), the gauge potential has to satisfy the anti-Hermiticity condition

$$(A_{\bar{z}}^u d\bar{z})^\dagger = -A_z^u dz. \tag{35}$$

On the other hand, in a holomorphic gauge (a local choice of a holomorphic basis of the fibre), we have

$$\bar{\partial}_{A^h} = \bar{\partial}. \tag{36}$$

It is now straightforward to check that any connection which is compatible with both the unitary and the holomorphic structure must have curvature of type $(1,1)$. This follows by a short computation which is important for us and which we therefore spell out. The gauge change from the holomorphic to the unitary gauge must be via a locally defined map $g : U \subset \Sigma \to GL(2, \mathbb{C})$ satisfying

$$\bar{\partial}_{A^u} = \bar{\partial} + g \bar{\partial} g^{-1}. \tag{37}$$

But then the condition (35) fixes the connection uniquely and implies an explicit formula for the gauge potential in the unitary gauge, valid in the open set $U \subset \Sigma$:

$$A^u = g \bar{\partial} g^{-1} + (g^{-1})^\dagger \partial g^\dagger. \tag{38}$$

This shows in particular that if $A^u$ is the gauge potential of an $SU(2)$ connection, then $g$ has determinant 1 and is therefore $SL(2, \mathbb{C})$-valued. This expression for a gauge potential in two dimensions is also frequently used in the physics literature on planar $SU(2)$ Yang-Mills theory, see e.g. [17].

We can transform back, using $g^{-1}$, from the unitary to the holomorphic gauge to deduce the $(1,0)$ component and hence the entire gauge field in the holomorphic gauge as

$$A^h = g^{-1}(g^{-1})^\dagger \partial g^\dagger g + g^{-1} \partial g = h^{-1} \partial h, \tag{39}$$

where we defined $h = g^\dagger g$. The matrix $h$ is manifestly Hermitian and positive definite, and defines a Hermitian inner product on the fibre in the holomorphic gauge [13]. The curvature 2-form then comes out as

$$F = \bar{\partial}(h^{-1} \partial h), \tag{40}$$

---

[1] Sections of a bundle $E$ over $\Sigma$ are holomorphic if they are holomorphic as maps from $\Sigma$ into the total space of the bundle $E$.

which is manifestly of type $(1, 1)$ (and will remain so after gauge transformations), as claimed.

One can also prove the converse result [12, 13]. If a complex vector bundle $E$ over a complex manifold, with a unitary structure, has a connection which is unitary and has curvature of type $(1, 1)$ then there is a unique holomorphic structure on $E$ such that the connection has the forms (38) and (39) in the unitary and holomorphic gauge respectively.

In one complex dimension, any connection has curvature of type $(1, 1)$ and it follows that the unitary connections on $\Sigma$ which we considered in Sect. 2 define complex structures on the total space of the $S^2$-bundle $P \times^{\mathrm{Ad}} S^2$ over $\Sigma$. Since we are interested in $SU(2)$ connections, they can always locally be expressed in the form (38) for $g : U \to SL(2, \mathbb{C})$. This is the result which we will put to practical use in the next section. As a final preparation we recall the Iwasawa decomposition of $g \in SL(2, \mathbb{C})$ via

$$g = u\rho, \tag{41}$$

with $u \in SU(2)$ and $\rho$ an upper-triangular matrix with unit determinant of the form

$$\rho = \begin{pmatrix} \lambda & c \\ 0 & \frac{1}{\lambda} \end{pmatrix}, \qquad \lambda \in \mathbb{R}^+, \, c \in \mathbb{C}. \tag{42}$$

Since the unitary factor $u$ acts as an overall unitary gauge transformation in (38) we can express any Hermitian gauge potential on $\Sigma$ up to $SU(2)$ gauge transformation locally as

$$A = \rho\bar{\partial}\rho^{-1} + (\rho^{-1})^\dagger \partial\rho^\dagger, \tag{43}$$

where $\rho$ is a matrix-valued function of the form (42).

## 3.2 Holomorphic structure of the gauged sigma model

In order to apply the theory of the previous section to the gauged sigma model of Sect. 2, we need a little more notation. We write vectors in $\mathbb{C}^2$ as

$$\vec{w} = \begin{pmatrix} w_1 \\ w_2 \end{pmatrix}, \tag{44}$$

and use the standard Hermitian product $\langle \vec{v}, \vec{w} \rangle = v_1\bar{w}_1 + v_2\bar{w}_2$ on $\mathbb{C}^2$. The Hopf projection maps the unit sphere $S^3$ in $\mathbb{C}^2$ to the unit sphere $S^2$ in the Lie algebra $su(2)$ via

$$\pi : S^3 \subset \mathbb{C}^2 \to S^2 \subset su(2), \quad \vec{w} \mapsto n = W t_3 W^{-1}, \qquad W = \begin{pmatrix} w_1 & -\bar{w}_2 \\ w_2 & \bar{w}_1 \end{pmatrix}, \tag{45}$$

or, with $n = n_1 t_1 + n_2 t_2 + n_3 t_3$,

$$n_1 + in_2 = 2w_2\bar{w}_1, \qquad n_3 = |w_1|^2 - |w_2|^2. \tag{46}$$

The standard action of $u \in SU(2)$ on $\mathbb{C}^2$,

$$u : \mathbb{C}^2 \to \mathbb{C}^2, \quad \vec{w} \mapsto u\vec{w}, \tag{47}$$

induces the adjoint action,

$$u : su(2) \to su(2), \qquad n \mapsto unu^{-1} = R(u)n, \tag{48}$$

which preserves the inner product (11) in $su(2)$.

We use the conventions of [7] to define a stereographic coordinate $w \in \mathbb{C} \cup \{\infty\}$ for the 2-sphere by projection from the south pole,

$$w = \mathrm{St}(n) = \frac{n_1 + in_2}{1 + n_3}, \tag{49}$$

with inverse

$$n_1 + in_2 = \frac{2w}{1 + |w|^2}, \quad n_3 = \frac{1 - |w|^2}{1 + |w|^2}. \tag{50}$$

One checks that the Hopf projection (45) followed by stereographic projection can now also be written as

$$\mathrm{St} \circ \pi : \vec{w} \mapsto w = \frac{w_2}{w_1}. \tag{51}$$

An element

$$g = \begin{pmatrix} a & b \\ c & d \end{pmatrix} \in SL(2, \mathbb{C}) \tag{52}$$

acts on $\vec{w} \in \mathbb{C}^2$ by ordinary matrix multiplication

$$g : \vec{w} \mapsto g\vec{w}, \tag{53}$$

and on our projective coordinate $w$ by fractional linear transformation, which we write as

$$w \mapsto g[w] := \frac{c + dw}{a + bw}. \tag{54}$$

For $u \in SU(2) \subset SL(2, \mathbb{C})$, this action agrees with (48) when $w$ and $n$ are related via the stereographic map (49). However, non-unitary elements in $SL(2, \mathbb{C})$ act as conformal transformations which do not preserve the round metric induced by the embedding $S^2 \subset su(2)$.

To write the gauged non-linear sigma model in terms of the stereographic coordinate $w$, we note that our $sl(2, \mathbb{C})$ Lie algebra generators (10) are explicitly

$$t_+ = t_1 + it_2 = \begin{pmatrix} 0 & -i \\ 0 & 0 \end{pmatrix}, \quad t_- = t_1 - it_2 = \begin{pmatrix} 0 & 0 \\ -i & 0 \end{pmatrix}, \quad t_3 = \begin{pmatrix} -\frac{i}{2} & 0 \\ 0 & \frac{i}{2} \end{pmatrix}. \tag{55}$$

Writing their action on the projective coordinate $w$ simply as juxtaposition, we have, for general $t \in sl(2, \mathbb{C})$,

$$tw = \left. \frac{d}{d\epsilon} \right|_{\epsilon=0} \exp(\epsilon t)[w], \quad t \in sl(2, \mathbb{C}), \tag{56}$$

and compute

$$t_- w = -i, \quad t_3 w = iw, \quad t_+ w = iw^2. \tag{57}$$

Defining the Lie algebra components (as opposed to the 1-form components (8)) of the gauge potential $A$ via

$$A = \frac{1}{2}(A_+ t_- + A_- t_+) + A_3 t_3, \tag{58}$$

and similarly for the curvature

$$F = \frac{1}{2}(F_+ t_- + F_- t_+) + F_3 t_3, \tag{59}$$

we can write the covariant derivative as

$$Dw = dw + Aw = dw - \frac{i}{2}A_+ + iA_3 w + \frac{i}{2}A_- w^2, \tag{60}$$

and have the identity

$$(F, n) = \frac{wF_- + \bar{w}F_+ + F_3(1 - |w|^2)}{1 + |w|^2}. \tag{61}$$

Using the standard expression for the Dirichlet term in terms of stereographic coordinates [16], the energy (12) of the gauged non-linear sigma model then takes the form

$$E[A, w] = \int_\Sigma 2 \frac{Dw \wedge \star \overline{Dw}}{(1 + |w|^2)^2} - \int_\Sigma \frac{wF_- + \bar{w}F_+ + F_3(1 - |w|^2)}{1 + |w|^2}, \tag{62}$$

and the identity (22) reads

$$2i \frac{Dw \wedge \overline{Dw}}{(1 + |w|^2)^2} = 2i \frac{dw \wedge \overline{dw}}{(1 + |w|^2)^2} + \frac{wF_- + \bar{w}F_+ + F_3(1 - |w|^2)}{1 + |w|^2}$$
$$- d\left( \frac{wA_- + \bar{w}A_+ + A_3(1 - |w|^2)}{1 + |w|^2} \right). \tag{63}$$

With

$$(Dw - i \star Dw) \wedge \star \overline{(Dw - i \star Dw)} = 2 Dw \wedge \star \overline{Dw} - 2i Dw \wedge \overline{Dw}, \tag{64}$$

the energy can be therefore be written as

$$E[A, w] = \int_\Sigma \frac{(Dw - i \star Dw) \wedge \star \overline{(Dw - i \star Dw)}}{(1 + |w|^2)^2} + 2i \int_\Sigma \frac{dw \wedge \overline{dw}}{(1 + |w|^2)^2}$$
$$- \int_{\partial \Sigma} \frac{wA_- + \bar{w}A_+ + A_3(1 - |w|^2)}{1 + |w|^2}. \tag{65}$$

The second term is $4\pi$ times the degree of $w$, and the last term is a boundary term. If both degree and boundary behaviour are kept fixed, minima of the energy are therefore determined by the equation

$$Dw = i \star Dw \Longleftrightarrow D_{\bar{z}} w = 0, \tag{66}$$

where we used the basic properties (3) of the $\star$-operator on 1-forms. This is the Bogomol'nyi equation (15) in stereographic coordinates, as can also be checked by explicitly changing coordinates according to (50) in (15).

A key feature of the equation (66), which was not obvious in the formulation (15), is its gauge invariance under the larger group of $SL(2, \mathbb{C})$-valued gauge transformation

$$A_{\bar{z}} \mapsto g A_{\bar{z}} g^{-1} + g \partial_{\bar{z}} g^{-1}, \qquad w \mapsto g[w], \tag{67}$$

where $g : U \subset \Sigma \to SL(2, \mathbb{C})$, and we used the notation (54) for fractional linear transformations.

## 3.3  A general solution

We can now apply the geometrical considerations of Sect. 3.1 to solve the Bogomol'nyi equation (66) for a given $su(2)$-connection $A$ on the principal bundle $P$ as follows. We consider the $\mathbb{C}^2$-bundle associated to $P$ via (47). By the results of Sect. 3.1, the connection $A$ defines a holomorphic structure on this bundle, and hence also on the associated projective $\mathbb{C}P^1$-bundle. Locally, we can go to a holomorphic gauge via an $SL(2, \mathbb{C})$ gauge transformation. In this gauge $\bar{\partial}_A = \bar{\partial}$, so that the Bogomol'nyi equation (66) can easily be solved.

Explicitly, this means that, for a given unitary connection $A$, we need to find a locally defined map

$$g : U \subset \Sigma \to SL(2, \mathbb{C}), \tag{68}$$

so that the anti-holomorphic component of $A$ is

$$A_{\bar{z}} = g \partial_{\bar{z}} g^{-1}. \tag{69}$$

Then the Bogomol'nyi equation (66) becomes simply

$$\partial_{\bar{z}} w + g \partial_{\bar{z}} g^{-1} w = 0. \tag{70}$$

Using again our notation (54) for the action of $g$ on $w$ by fractional linear transformations, this means that $f = g^{-1}[w]$ is a holomorphic function. Thus we obtain the general solution, valid in some open set $U \subset \Sigma$, as

$$w = g[f], \quad \text{with} \quad f : U \to \mathbb{C}P^1 \quad \text{holomorphic}. \tag{71}$$

The field $n$ can then be reconstructed via (50). We will illustrate this formalism in the next section by applying it to models of magnetic skyrmions, written as gauged non-linear sigma models.

# 4 Applications to magnetic skyrmions and impurities

## 4.1 Critically coupled magnetic skyrmions with any DM term

In the remainder of the paper we focus on models defined in the plane $\Sigma = \mathbb{C}$. For our solution of the gauged sigma model, both holomorphic and unitary gauges can be chosen globally in this case and therefore we obtain a global solution of the form (71).

In order to apply our method to magnetic skyrmions we consider the general model already presented in the Introduction, but now write the energy (1) in Lie algebraic notation, which simply means interpreting the magnetisation vector $\boldsymbol{n}$ as a unit length element $n$ of the Lie algebra $su(2)$, and writing vector products as commutators:

$$E_S[n] = \int_{\mathbb{R}^2} \frac{1}{2}|\partial_1 n|^2 + \frac{1}{2}|\partial_2 n|^2 + \sum_{a=1}^{3}\sum_{i=1}^{2} \mathcal{D}_{ai}[\partial_i n, n]_a + V(n) \, dx_1 dx_2. \tag{72}$$

Here the index $a$ refers to the components with respect to Lie algebra basis $t_a$ introduced in Sect. 2.1, so

$$[\partial_i n, n]_a = (t_a, [\partial_i n, n]). \tag{73}$$

In order to have a translation-invariant theory, we assume that the spiralization tensor $\mathcal{D}$ is constant.

We now write the model (72) as a gauged non-linear sigma model for a translation-invariant $SU(2)$ gauge potential

$$A = A_1 dx_1 + A_2 dx_2, \tag{74}$$

where $A_1, A_2$ are Lie-algebra valued constants. Noting that the curvature is

$$F = [A_1, A_2] \, dx_1 \wedge dx_2, \tag{75}$$

the energy functional (13) of the gauged non-linear sigma model takes the form

$$E[A, n] = \int_{\mathbb{R}^2} \left( \frac{1}{2}|\partial_1 n|^2 + \frac{1}{2}|\partial_2 n|^2 - (A_1, [\partial_1 n, n]) - (A_2, [\partial_2 n, n]) \right.$$
$$\left. + \frac{1}{2}|[A_1, n]|^2 + \frac{1}{2}|[A_2, n]|^2 - (n, [A_1, A_2]) \right) dx_1 dx_2. \tag{76}$$

The key step in the translation from magnetic skyrmions to gauged non-linear sigma models is the identification of the gauge field with the spiralization tensor according to

$$A_i = -\sum_{a=1}^{3} \mathcal{D}_{ai} t_a, \quad i = 1, 2. \tag{77}$$

In other words, the spiralization tensor is interpreted as minus the matrix obtained when expanding $A_1, A_2$ in the basis $t_a$ of $su(2)$. This prescription should be viewed as special case of the more natural three-dimensional situation, where we have a further component $A_3$ of the gauge field and the spiralization tensor is minus the $3 \times 3$ matrix representing the linear map which takes the basis elements $t_a$ into $A_i$, $i = 1, 2, 3$.

The DM term can then be written in terms of the gauge field as

$$\sum_{a=1}^{3} \sum_{i=1}^{2} \mathcal{D}_{ai} [\partial_i n, n]_a = -\sum_{i=1}^{2} (A_i, [\partial_i n, n]). \tag{78}$$

If we now pick the potential

$$V_A(n) = \frac{1}{2} |[A_1, n]|^2 + \frac{1}{2} |[A_2, n]|^2 - (n, [A_1, A_2]), \tag{79}$$

then the energy functional (72) for magnetic skyrmions with the potential $V = V_A$ equals the expression (76) with the particular gauge field (77). This observation is one of the key results of this paper, and allows us to obtain stationary points of the magnetic skyrmion energy (72) by solving the Bogomol'nyi equation (66) for the gauged non-linear sigma model with gauge field (77). The choice of potential (79) for any given spiralization tensor generalises the notion of 'critically coupled' introduced [7], and we will use this term to describe the solvable models of magnetic skyrmions defined by (72) with $V = V_A$.

Solving the Bogomol'nyi equations turns out to be straightforward. Since

$$A_{\bar{z}} = \frac{1}{2} (A_1 + i A_2) = g \partial_{\bar{z}} g^{-1}, \tag{80}$$

for constant $A_1$ and $A_2$, is solved by

$$g = \exp(-\frac{1}{2} (A_1 + i A_2) \bar{z}), \tag{81}$$

we obtain the general solution from (71). Since $A_{\bar{z}}$ is generally a complex, traceless $2 \times 2$ matrix, the explicit form of $g$ as a $2 \times 2$ matrix can be calculated by noting that $A_{\bar{z}}$ is conjugate (by a $SL(2, \mathbb{C})$ matrix) either to a diagonal matrix with equal and opposite complex eigenvalues (the generic case) or to a nilpotent matrix. We consider and interpret the special cases where $A_{\bar{z}}$ is nilpotent or has purely imaginary eigenvalues in some detail below.

Regarding the general case, we note that, if

$$A_{\bar{z}} = \frac{1}{2} \begin{pmatrix} \lambda + i\omega & 0 \\ 0 & -\lambda - i\omega \end{pmatrix}, \tag{82}$$

then the general solution (71) is

$$w = e^{(\lambda + i\omega)\bar{z}} f(z), \tag{83}$$

for some holomorphic map $f$. If $A_{\bar{z}}$ is conjugate to (82) via a constant $h \in SL(2, \mathbb{C})$ then the corresponding solution is obtained by acting with $h$ on (83) via fractional linear transformations according to (54); the magnetisation field $n$ is obtained via (49) as before.

## 4.2 Axisymmetric DM interactions

The Dirichlet energy term in (72) is invariant under translations, reflections and rotations in the plane and under reflections and rotations (about any axis) of the magnetisation field $n$. The DM interaction breaks this symmetry, and for generic but constant spiralization tensors the breaking is maximal, leaving only the translational symmetry intact. However, for particular choices of $\mathcal{D}$, the DM term is invariant under rotations and reflections in the plane and simultaneous rotations and reflections of the magnetisation vector $n$. The symmetry group is isomorphic to $O(2)$ and we call such DM terms axisymmetric. They are easily characterised in terms of our gauge potential $A$. The DM term is axisymmetric if and only if a spatial rotation of the gauge field

$$A_1 \mapsto \cos\beta\, A_1 + \sin\beta\, A_2, \qquad A_2 \mapsto -\sin\beta\, A_1 + \cos\beta\, A_2, \tag{84}$$

can be written as a rotation in the $su(2)$ Lie algebra, i.e., by a conjugation of $A_1$ and $A_2$ with a $SU(2)$ matrix. This is the case if and only if

$$|A_1|^2 = |A_2|^2, \qquad (A_1, A_2) = 0, \tag{85}$$

i.e. if $A_1, A_2$ and $[A_1, A_2]$ form an, up to scale, orthonormal basis of $su(2)$. In this case, there exists a rescaling by $\kappa > 0$ and a $SU(2)$ matrix $u$ so that

$$\kappa u t_1 u^{-1} = A_1, \quad \kappa u t_2 u^{-1} = A_2, \quad \kappa^2 u t_3 u^{-1} = [A_1, A_2]. \tag{86}$$

The DM term is then invariant under spatial rotations (84) and the simultaneous rotation of the magnetisation vector according to

$$n \mapsto R(u) R_3(\beta) R(u)^{-1} n, \tag{87}$$

where $R(u)$ is the $SO(3)$ matrix associated to the $SU(2)$ matrix $u$ via (48) and $R_3(\beta)$ is the rotation about $t_3$ by $\beta$. It is also invariant under reflections

$$A_1 \mapsto A_1, \qquad A_2 \mapsto -A_2, \qquad n \mapsto R(u) S_{13} R(u)^{-1} n, \tag{88}$$

where $S_{13}$ is the reflection in the 13 plane. Note also that the potential (79) takes a particularly simple form when (85) holds:

$$V_A(n) = \frac{1}{2\kappa^2} \left( \kappa^2 - (n, [A_1, A_2]) \right)^2. \tag{89}$$

The condition (85) also leads to a considerable simplification in the solution of the model. It is easy to check that it is equivalent to the matrix $A_{\bar{z}}$ being nilpotent and therefore conjugate (via the $SU(2)$ matrix $u$) to a matrix of the form

$$\begin{pmatrix} 0 & * \\ 0 & 0 \end{pmatrix}, \tag{90}$$

for some complex entry $*$. This is the case considered in [7], which dealt with the DM term

$$\sum_{a=1}^{3} \sum_{i=1}^{2} \mathcal{D}_{ai} [\partial_i n, n]_a = \kappa \cos\alpha\, w_B + \kappa \sin\alpha\, w_N, \tag{91}$$

where

$$\begin{aligned} w_B &= n_1 \partial_2 n_3 - n_2 \partial_1 n_3 + n_3 (\partial_1 n_2 - \partial_2 n_1), \\ w_N &= -n_1 \partial_1 n_3 + n_2 \partial_2 n_3 + n_3 (\partial_1 n_1 + \partial_2 n_2). \end{aligned} \tag{92}$$

In our notation this corresponds to the spiralization tensor

$$\mathcal{D} = \kappa \begin{pmatrix} \cos\alpha & \sin\alpha \\ -\sin\alpha & \cos\alpha \\ 0 & 0 \end{pmatrix}, \tag{93}$$

and hence

$$A_1 = -(\cos\alpha\, t_1 - \sin\alpha\, t_2) \quad A_2 = -(\sin\alpha\, t_1 + \cos\alpha\, t_2), \tag{94}$$

or, with the conventions (55),

$$A_{\bar{z}} = -\frac{1}{2}\kappa e^{i\alpha} t_+ = \begin{pmatrix} 0 & \frac{i}{2}\kappa e^{i\alpha} \\ 0 & 0 \end{pmatrix}. \tag{95}$$

In this case the potential is simply

$$V_A(n) = \frac{\kappa^2}{2}(1-n_3)^2. \tag{96}$$

To apply our method of solution, we note that

$$g = \exp\left(\frac{\kappa}{2}e^{i\alpha}\bar{z}t_+\right) = \begin{pmatrix} 1 & -\frac{i}{2}\kappa e^{i\alpha}\bar{z} \\ 0 & 1 \end{pmatrix} \tag{97}$$

solves (69) for the gauge field (95). Thus, the general solution (71) is given in terms of a holomorphic function $f : \mathbb{C} \to \mathbb{C}P^1$ and the fractional linear transformation (54) as

$$w = g[f] = \frac{f}{1 - \frac{i}{2}\kappa e^{i\alpha}\bar{z}f}. \tag{98}$$

In terms of $v = 1/w$ this is

$$v = -\frac{i}{2}\kappa e^{i\alpha}\bar{z} + \frac{1}{f}, \tag{99}$$

which, after re-naming $f \to 1/f$, is the general solution found and discussed in [7].

In the more general case (85), the solutions of the model are obtained from solutions (98) by rotating with $R(u)$. However, the physics is quite different. Even if both $A_1$ and $A_2$ are in the $t_1 t_2$-plane, the two possible directions of $[A_1, A_2]$ lead to different chiralities [6,7]. In the generic case, we obtain a linear combination of 'in plane' and 'out of plane' DM terms. This illustrates that a rotation, which is a change of gauge in the gauged sigma model, can make a physical difference in the interpretation as a model of magnetic skyrmions.

## 4.3 Rank one DM interaction

The spiralization matrix $\mathcal{D}$, still assumed to be constant, has rank 1 when $A_1$ and $A_2$ are constant and collinear, so when

$$[A_1, A_2] = 0. \tag{100}$$

In this case, the curvature $F$ of $A$ vanishes. As a result, the gauge field can be removed entirely by a gauge transformation. The solutions are then related to the Belavin-Polyakov solitons of the $O(3)$ sigma model [8] (holomorphic or anti-holomorphic maps $\Sigma \to \mathbb{C}P^1$) by space-dependent $SU(2)$ transformations. Our general formula (71) reproduces the solutions related to holomorphic maps. For example

$$A_1 = a t_3, \qquad A_2 = A_3 = 0, \qquad a \in \mathbb{R}, \tag{101}$$

leads to the potential

$$V_A(n) = \frac{a^2}{2}(n_1^2 + n_2^2) = \frac{a^2}{2}(1 - n_3^2),\tag{102}$$

and the general solution

$$w = e^{-\frac{i}{2}a\bar{z}}f(z),\tag{103}$$

where $f$ an arbitrary holomorphic function. The simplest case is $f = 1$, and produces the configuration

$$n = \begin{pmatrix} \operatorname{sech}\left(\frac{a}{2}x_2\right)\cos\left(\frac{a}{2}x_1\right) \\ -\operatorname{sech}\left(\frac{a}{2}x_2\right)\sin\left(\frac{a}{2}x_1\right) \\ \tanh\left(\frac{a}{2}x_2\right) \end{pmatrix}.\tag{104}$$

This is a kink interpolating between the 'up' and the 'down' vacuum of the potential in the $x_2$-direction and rotating in the $x_1$-direction at the same time. One can get rid of the kink (leading to a helix) or of the rotation (leading to a pure kink) by choosing

$$f(z) = e^{-i\frac{a}{2}z} \quad \text{or} \quad f(z) = e^{i\frac{a}{2}z}.\tag{105}$$

If one picks

$$f(z) = e^{-i\frac{a}{2}z}r_n(z),\tag{106}$$

where $r_n$ is a rational map of degree $n$, one obtains

$$w = e^{-iax_1}r_n(z),\tag{107}$$

i.e. a rational map modulated by a helix in the $x_1$-direction. This describes a Belavin-Polyakov multi-soliton [8] modulated by a helix.

### 4.4 Impurities as non-abelian gauge fields

To end our discussion of applications, we indicate how the interaction of Belavin-Polyakov solitons with impurities can also be described in terms of the gauged non-linear sigma model introduced in this paper. Unlike in the application to magnetic skyrmions, the non-abelian connection will need a non-trivial spatial variation to model an impurity.

In the recent paper [18], the authors study mathematical models for Belavin-Polyakov solitons which interact with impurities in a way which preserves supersymmetry. The first order equation for solitons in the presence of an impurity proposed in that paper is rather similar to (66), but with a given impurity configuration instead of a gauge field. We will now derive the impurity equation of [18] from our Bogomol'nyi equation (66), explain how to pick the gauge field to reproduce the impurities studied in [18] and point out generalisations.

Extending the discussion of [18] to an arbitrary Riemann surface, and using the complex notation introduced in Sect. 3.2 to parametrise $\mathbb{C}P^1$ in terms of $\mathbb{C} \cup \{\infty\}$, the impurity field considered in [18] is a map

$$\sigma : \Sigma \to \mathbb{C}P^1.\tag{108}$$

Further adapting conventions so that the soliton field $u$ considered in [18] is our field $\bar{w}$, the simplest equation of [18] for a configuration $w$ coupled to such an impurity is

$$\partial_{\bar{z}}w + \sigma = 0.\tag{109}$$

To see that this is a special case of our Bogomol'nyi equation (66), we use the expansion (60) to write (66) as

$$\partial_{\bar{z}}w - \frac{i}{2}(A_+)_{\bar{z}} + i(A_3)_{\bar{z}}w + \frac{i}{2}(A_-)_{\bar{z}}w^2 = 0,\tag{110}$$

where we remind the reader that the indices $+, -, 3$ refer to the Lie algebra components (58). To obtain (109) we simply need to pick

$$-\frac{i}{2}(A_+)_{\bar{z}} = \sigma, \quad (A_3)_{\bar{z}} = \frac{i}{2}(A_-)_{\bar{z}} = 0, \tag{111}$$

which amounts to

$$A_{\bar{z}} = \begin{pmatrix} 0 & 0 \\ \sigma & 0 \end{pmatrix}, \qquad A_z = \begin{pmatrix} 0 & -\bar{\sigma} \\ 0 & 0 \end{pmatrix}. \tag{112}$$

One checks that this connection can be expressed according to (69) in terms of the $SL(2, \mathbb{C})$-valued map

$$g = \begin{pmatrix} 1 & 0 \\ -\int \sigma \, d\bar{z} & 1 \end{pmatrix}, \tag{113}$$

so that our general solution (71) in this case is

$$w = f - \int \sigma \, d\bar{z}, \tag{114}$$

with $f$ again holomorphic. This is the obviously the general solution of (109), and was also derived and studied in [18]. However, as also discussed in [18], simple properties like total energy and degree of configurations like (114) are rather subtle, despite the explicit form, echoing a similar observation regarding exact magnetic skyrmions in [7].

The authors of [18] also discuss Bogomol'nyi equations for solitons interacting with impurities which contain products of the impurity field and the soliton field. In our general equation (110) this would simply amount to picking a connection $A$ with a non-trivial component $A_3$. Our general solution again applies. The possibility of interactions which are quadratic in the soliton field correspond to a non-trivial component $A_-$ of the gauge field. This does not appear to have been considered in the literature on impurities, but would seem equally natural from our point of view.

# 5 Conclusion

In this paper we introduced and studied the gauged non-linear sigma model defined by the energy functional (12). We showed that it provides a framework for systematically studying solvable theories of magnetic skyrmions with any given DM interaction term. We also indicated that it provides a natural language for solvable theories of Belavin-Polyakov solitons interacting with impurities.

We showed how to solve the first order Bogomol'nyi equation (66) of the gauged non-linear sigma model by exploiting the relation between holomorphic and unitary gauges. This leads to the explicit formula (71) for local solutions. In the application considered here we assumed $\Sigma = \mathbb{C}$, in which case we obtain infinitely many globally defined magnetic skyrmions for any given spiralization tensor, and similarly infinitely many globally defined solitons in the presence of any given impurity.

Summed up as a slogan, our results show that the geometry behind both DM interactions and impurities in the $O(3)$ sigma model is a Chern connection. They also suggest that DM interactions and impurities may be different aspects of the same underlying physics, and that tools used in the study of one may be usefully applied to the other. For example, the supersymmetric nature of the model for impurities studied in [18] suggests that the non-abelian sigma model studied here also has a natural supersymmetric extension. This may, in turn, have interesting implications for magnetic skyrmions in the theories we studied here.

From a mathematical, and possibly also physical, point of view it would certainly be of interest to repeat our analysis on more general Riemann surfaces, and to study global properties

of solutions (71) there. This requires a choice of unitary connection. As we explained, such connections define complex structures on $\mathbb{C}^2$-bundles and hence on associated $\mathbb{C}P^1$-bundles over such Riemann surfaces, and this provides a natural geometrical interpretation for any chosen connection.

It would also be interesting to consider Riemann surfaces with boundary, and to clarify the correct boundary conditions in each case. We only briefly touched on the relevant issues in Sect. 2.4 of this paper.

## Acknowledgements

Some of the results in this paper were presented in two seminars I gave in Edinburgh in early 2019. I thank members of the audience for insightful comments, Lorenzo Foscolo for several discussions and Calum Ross for comments on an earlier version of this manuscript. I also thank the SciPost referees for constructive comments which helped improve the presentation.

## Note added in proof

During the proof stage of this paper, the e-print [19] appeared on the arxiv in which the author points out that the gauged sigma model and the Bogomol'nyi equation proposed here can naturally be expressed and generalised in the language of $J$-holomorphic curves and equivariant cohomology, as discussed in [20]

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
