# Peer review of "Gauged Sigma Models and Magnetic Skyrmions"

_SciPost Physics, doi:SciPost Phys. 7, 030 (2019)_

## Round 1 · Referee Report · Anonymous (Referee 1) · 2019-6-11

Strengths

The paper constructs a novel energy functional in which a gauge constraint ensures that all allowed configurations extremize the energy.

Weaknesses

It's not clear what the applications of this energy functional are, beyond those pointed out in the previous paper.

The paper is written in a differential geometric language that does little to improve the rigor of the work but makes the paper difficult to read for the condensed matter audience.

Report

The paper studies a class of solitons in field theories in two spatial dimensions. Among these solitons are magnetic Skyrmions, which are of some interest in condensed matter systems.

The paper writes down an energy functional for an SU(2) gauge field coupled to an adjoint scalar. The gauge field appears in the energy only algebraically and so its equation of motion gives rise to a constraint on the allowed field configurations. The energy functional has the peculiar property that all solutions of this constraint also obey the equations of motion of the scalar field. This means that all solutions to the constraint are automatically extrema of the action. The configurations can be labelled by a topological charge and this determines the energy.

This is an interesting observation. But it's not clear to me what can be done with it. The set of all field configurations that obey the constraint appear to be too large to be of interest. Instead, the author suggests that one should not view the gauge field as dynamical, but instead fix it to some particular configuration. With this interpretation, the constraint need not be imposed but can instead be interpreted as a Bogomoln'yi equation of a related field theory. This is in keeping with a previous paper by the author, listed as [1] in the bibliography, in which it was pointed out that, for a particular choice of this gauge field, the model reduces to a model for magnetic Skymions, tuned to a BPS limit.

This suggests that the model could be viewed as unifying system, in which different choices of gauge field give different theories. But no other interesting model is introduced in this way. There is a suggestion in the conclusions that this may be useful to study BPS impurities, but this is not pursued. A general solution to the constraint equation is given, but the main application seems to be a recapitulation of the results of [1].

In summary, there is an interesting observation in this paper. However, the implications of this observation remain unclear, and suggestions to find a novel application are not followed through. For this reason, I do not think that this paper contains enough material to warrant publication in SciPost.
  • validity: good
  • significance: ok
  • originality: ok
  • clarity: good
  • formatting: perfect
  • grammar: perfect

Author:  Bernd Schroers  on 2019-06-13  [id 539]

(in reply to Report 1 on 2019-06-11)
Category:
objection

I thank the referee for the careful report. However, I think it does not properly take into account that this is an interdisciplinary paper which does two things:

  1. The paper provides a (new, as far as I know) geometrical framework which explains the results of paper [1] in terms of complex geometry. This allows for a vast generalisation of the results of [1], but, more importantly from the point of view of mathematical physics, provides a geometrical understanding of them: critically coupled magnetic skyrmions are holomorphic sections with respect to a connection defined by the DMI term! This is largely a mathematical advance, but experience shows that this kind of insight is also likely to be useful in applications.

  2. The paper briefly illustrates the power of the new framework with applications to magnetic skyrmions and defects.

If feel that the referee judges the merit of the paper almost entirely on its usefulness, i.e. point 2. I do not think is is a fair assessment, but would be able to do my part for balancing the two aspects as follows. While I think that a full discussion of applications would best be done in a separate paper which emphasises the physics and downplayed the mathematics, I would be happy to include further comments on other applications (beyond the one of paper [1]) which are currently only sketched in equations 3.47 ff. In particular, this would include comments on in-plane magnets and bi-merons.

---

## Round 1 · Referee Report · Anonymous (Referee 2) · 2019-7-5

Report

This is a fine paper that will be useful to those working on the mathematical formulation and solution of magnetic Skyrmion models. It also has ramifications beyond these Skyrmions. The paper extends a formalism introduced by the author and collaborators for such Skyrmions in a plane to more general Riemann surfaces. The main idea is to introduce a gauge potential, hence the title of the paper. The main result is to obtain a solvable first-order Bogomolny equation in this context.
Suggested minor improvement: Clarify what happens to the D_z n term in (2.14). Why does this disappear going to (2.15)?
Typos: skyrmions (p.1 near start). We think (p.2). term (p.5). which we will (p.7). Maybe delete "to" in "to (3.39)" (p.12).

---

## Round 1 · Referee Report · Anonymous (Referee 3) · 2019-7-8

Strengths

The mains strength of this paper is to provide a mathematical description of solutions of a gauged nonlinear sigma model on an arbitrary Riemann surface using standard BPS arguments.

Weaknesses

The case relevant for magnetic skyrmions is only sketched.

Report

A large part of the paper is devoted to an analysis of a nonlinear sigma model (NLSM) by using a map to an SU(2) gauge theory. By itself this type of approach is known for decades and has been considered by the author himself back in 1995 [B. J. Schroers, "Bogomol’nyi solitons in a gauged O(3) sigma model", Phys. Lett. B 356, 291 (1995); https://doi.org/10.101/0370-2693(95)00833-7]. In this sense the starting point of this paper is far from being innovative. One can basically find the same equations in the mentioned paper, except for the use of differential forms, which is more appropriate for discussing general Riemannian manifolds. Thus, it seems to me that one valid criticism would be whether this submission to SciPost is sufficiently novel to justify its publication. My first impression regarding this point is that this paper constitutes mostly a more mathematically general and precise extension of the above reference by the same author. The application to magnetic skyrmions, announced in the title, abstract, and introduction, is only slightly addressed, leaving the reader somewhat frustrated. In this regard we note, for instance, that after writing the energy functional (1.1), which contains a Dzyaloshinskii-Moriya (DM) term, the author proceeds with analyzing the energy functional (2.11) [or, equivalently, its coordinate representation, Eq. (2.12)], which does not contain the DM term. When the author finally comes to the application to magnetic skyrmions in Section 3.3, we realize that most of it corresponds to a reformulation of results previously discussed in Ref. 1 of the paper, which is co-authored by the author of this SciPost submission. The author also advertises the system with impurities as a possible application of the method. It would have been nicer if the author were more specific here.

My next point touches the presentation of the mathematical formalism. Although the mathematical language of forms chosen by the author is appropriate, it is sometimes used in a way that masks obvious facts. For instance, let us consider the last term in Eq. (2.12), corresponding to the coupling of the field strength to the n field. It is easy to see that this term is just the topological charge. The author could have started the discussion by telling the significance of this term from the outset. Within a point of view more akin to Ref. 1, one writes $\vec{A}_\mu=\vec{n}\times\partial_\mu\vec{n}$, which immediately yields $\vec{F}_{12}=2(\partial_1\vec{n}\times\partial_2\vec{n})+\vec{n}\vec{n}\cdot(\partial_1\vec{n}\times\partial_2\vec{n})$, and therefore, $\vec{F}_{12}\cdot\vec{n}=3\vec{n}\cdot(\partial_1\vec{n}\times\partial_2\vec{n})$, being in this way proportional to the topological charge density. Thus, the last term in (2.12) is related to Eq. (2.20). I think it would help to alert the reader for this fact in a more explicit way than it is currently done in the paper.

Another point that it is maybe worth addressing (optional) and possibly relates to the point of gauge choices, is that the covariant derivative terms in (2.12) can also be written as something proportional to $\vec{A}^2$, assuming that $\vec{A}_\mu=\vec{n}\times\partial_\mu\vec{n}$. This would make the energy functional look like one corresponding to a massive gauge field at infinitely strong gauge coupling. I remember that such type of functional is associated to an infinite number of conserved currents. Does this fact have any impact for theory discussed in the paper?

Summarizing, I do not think that this paper should be published in its current form. However, I believe that the paper can be reconsidered if the author manages to improve the discussion and give more substance to the application to magnetic skyrmions.

Requested changes

The possible changes can be read off directly from the report. Being more specific, the most crucial changes would be:

1- Clearly state what are the main novelties (besides considering the model on a Riemann surface) relative to Ref. 1 and to the 1995 paper by the author mentioned in the report.

2- Improve and extend the discussion on the applications to magnetic skyrmions. It would be desirable to be more concrete with respect to the system in the presence of impurities.

  • validity: high
  • significance: low
  • originality: low
  • clarity: ok
  • formatting: perfect
  • grammar: perfect

Author:  Bernd Schroers  on 2019-07-15  [id 559]

(in reply to Report 3 on 2019-07-08)

I thank the referee for the comments, but I am puzzled by the report, which reflects a misunderstanding of the paper in key parts.

  1. The gauge field considered here is non-abelian and the model is therefore totally different from the abelian model I considered in my 1995 paper. The energy functionals are also different. The energy functional I consider here is new and my paper makes the case that it is important mathematically (its critical points are holomorphic sections of non-trivial $CP^1$ bundles over Riemann surfaces) and physically (it reproduces all DMI terms which have been considered in the literature on magnetic skyrmions).

  2. Contrary to what the referee says, the DMI terms is contained in the energy expression 2.12 from the start - it is the cross term of the gauge field with the derivative term, as show in 3.40.

  3. The gauge field $A_\mu = n\times \partial_\mu n$ which the referee mentions is commented on in equation 2.26 as [n,dn]. But this is only a side issue. As I stress from Sect. 3 onwards, my solution applies for any given non-abelian gauge field. The simple calculation suggested by the referee therefore only deals with a very special case.

I am happy to take the comments on board by providing more guidance to the reader, in particular by alerting the reader to the DMI discussion in Section 4, and extending that discussion.

Anonymous on 2019-07-15  [id 560]

(in reply to Bernd Schroers on 2019-07-15 [id 559])

Referee of report 3 here. Thank you for the clarifications. However, let me briefly add one more remark to one of my comments, which may help to shape the revised version.

Indeed, in your 1995 paper the gauge field is Abelian. However, the covariant derivative in Eq. (2.6) of that paper looks similar to one of a non-Abelian O(3) theory. The actual non-Abelian case would have a gauge field of the form $\vec{A}_\mu=A_\mu\vec{n}+\vec{n}\times\partial_\mu\vec{n}$, akin to 't Hooft's solution for the monopole. The Abelian character follows because the second term is absent. Maybe it would be useful if you cite your own 1995 paper as well and emphasize the differences between both approaches (Abelian Vs. non-Abelian). This can be done in the Introduction and eventually be mentioned in key passages of the text.

---

## Round 3 · Referee Report · Anonymous (Referee 2) · 2019-8-8

Report

The author has adopted the minor changes I recommended in my first report

---

## Round 3 · Referee Report · Anonymous (Referee 1) · 2019-8-8

Report

The revised version includes a substantial addition to the paper. This goes a long way towards alleviating my previous concerns that the paper did not contain enough novel material. I am happy to now recommend publication.

---

## Round 3 · Referee Report · Anonymous (Referee 3) · 2019-8-13

Report

The author addressed all the points raised by the referees. At least from my side there are no further concerns. I recommend the publication of the revised version in SciPost.

---

## Round 3 · Author Response

In response to the comments by referees 1 and 3 I have made major changes to the paper, detailed and justified below. I hope that they address the referees' concerns. I thank referee 2 for their positive remarks, and for pointing out typos, which I corrected.

---

## Round 3 · List of Changes

1. The introduction is substantially re-written and extended to bring out clearly that this paper goes far beyond the earlier paper [8] (in the new ordering of references), both in terms of applications (in deals with the most general DM interaction for magnetic skyrmions and includes impurities) and in terms of the underlying geometry (the underlying mathematical reason for solvability was not clear in [8] but is fully understood in this paper). The difference between this model and gauged non-linear sigma models studied elsewhere in the literature is also clarified.

  2. I have made minor changes to Sections 2 (some more details on boundary terms in 2.4) and 3 (a new final section 3.3 to highlight the general formula for solutions). Despite the reservations the referees have expressed about the use of differential forms I have not replaced them. I find them by far the most efficient tools for the calculations in these sections, and rely on them to bring out the mathematical structures which underpin the model. This would be harder and less clear in traditional vector calculus notation. I would also make the case that this paper has a strong interdisciplinary element, connecting condensed matter physics with differential geometry and mathematical physics, and that some compromise in the notation is therefore inevitable. Finally, I have made only very minor use of differential forms in the new Section 4 which deals with applications.

  3. I have introduced an entirely new Section 4 which spells out the application to magnetic skyrmions and impurities in much more detail then the previous version. This should bring out clearly the range of applications and also the extent to which the gauged sigma models provide a unifying picture for magnetic skyrmions and impurities.

  4. The new version uses the SciPost style file, and this automatically adds a table of contents.

---

## Editorial Decision

published